# Self-Forcing++: Towards Minute-Scale High-Quality Video Generation

**Justin Cui**[1,3]**, Jie Wu**[3,†]**, Ming Li**[2,3]**, Tao Yang**[3]**, Xiaojie Li**[3]**,
Rui Wang**[3]**, Andrew Bai**[1]**, Yuanhao Ban**[1]**, Cho-Jui Hsieh**[1,‡]

[1] UCLA [2] University of Central Florida [3] ByteDance Seed
† Project Leader, ‡ Corresponding Author

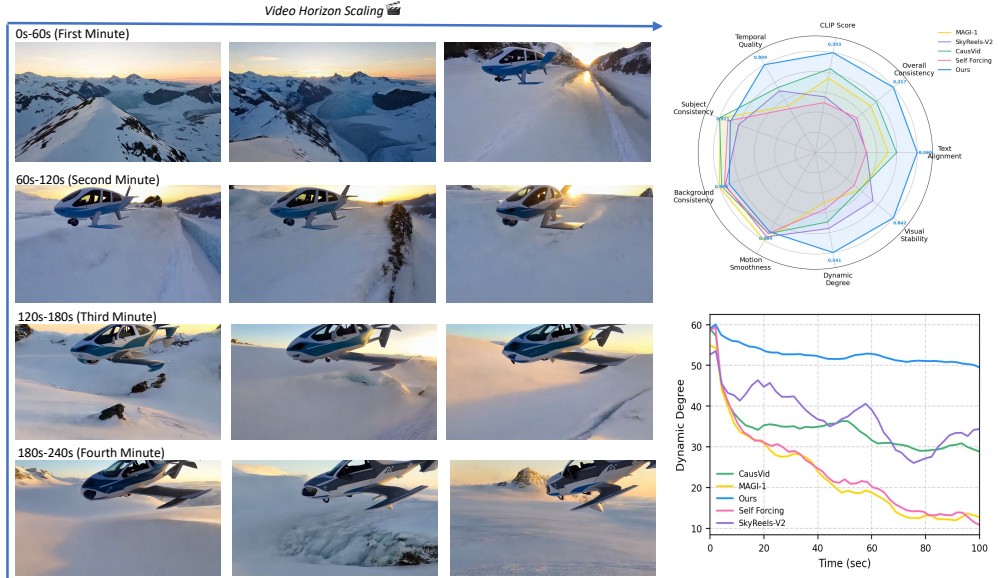

Figure 1: Self-forcing++ generates videos up to four minutes long. The radar chart highlights our model's superiority, while the line plot shows its sustained motion dynamics over long durations.

## Abstract

Diffusion models have revolutionized image and video generation, achieving unprecedented visual quality. However, their reliance on transformer architectures incurs prohibitively high computational costs, particularly when extending generation to long videos. Recent work has explored autoregressive formulations for long video generation, typically by distilling from short-horizon bidirectional teachers. Nevertheless, given that teacher models cannot synthesize long videos, the extrapolation of student models beyond their training horizon often leads to pronounced quality degradation, arising from the compounding of errors within the continuous latent space. In this paper, we propose a simple yet effective approach to mitigate quality degradation in long-horizon video generation without requiring supervision from long-video teachers or retraining on long video datasets. Our approach centers on exploiting the rich knowledge of teacher models to provide guidance for the student model through sampled segments drawn from self-generated long videos. Our method maintains temporal consistency while scaling video length by up to 20× beyond teacher's capability, avoiding common issues such as over-exposure and error-accumulation without recomputing overlapping frames like previous methods. When scaling up the computation, our method shows the capability of generating videos up to 4 minutes and 15 seconds, equivalent to 99.9% of the maximum span supported by our base model's position embedding and more than 50x longer than that of our baseline model. Experiments on standard benchmarks and our proposed improved benchmark demonstrate that our approach substantially outperforms baseline methods

in both fidelity and consistency. Our long-horizon videos demo can be found at self-forcing-plus-plus.github.io/.

# 1 INTRODUCTION

The field of video generation is advancing at a remarkable pace, catalyzed by the advent of diffusion models. Seminal works such as Sora (OpenAI, 2024), Wan (Wan et al., 2025), Hunyuan-DiT (Kong et al., 2024), and Veo (Google DeepMind, 2025) are progressively closing the gap between generated content and reality. Despite this progress, a formidable challenge remains: the majority of state-of-the-art models are confined to generating short-form videos, typically capped at 5-10 seconds. This constraint is inherent to the architectural design of the underlying Diffusion Transformers (DiT) (Peebles & Xie, 2023), the inherently non-streaming and non-causal nature of the vanilla DiT architecture poses a significant challenge to achieving temporal scalability.

A promising avenue for transcending this limitation lies in shifting from bidirectional diffusion architectures to autoregressive, streaming-based models. One such approach, Diffusion Forcing (Chen et al., 2024; Kim et al., 2024), applies heterogeneous noise schedules across frames to enable sequential generation. However, the combinatorial complexity of noise scheduling often leads to training instability and has proven difficult to scale (Chen et al., 2025; Sand-AI, 2025; He et al., 2025). A more tractable strategy involves predicting the next frame or chunk from a clean context, with KV caching emerging as a key mechanism for enabling performant, real-time streaming. For instance, CausVid (Yin et al., 2025) proposes a method to distill a bidirectional teacher model into a streaming student model using heterogeneous distillation. However, its reliance on overlapping frames for temporal consistency and a pronounced train-inference mismatch often results in over-exposure artifacts. The Self-Forcing (Huang et al., 2025) method mitigates the over-exposure issue by aligning the training and inference distributions. While this sets a new benchmark for short-form video quality, its capacity remains bottlenecked by the fixed-duration teacher model. Consequently, when tasked with generating content beyond this intrinsic temporal window (e.g., >10 seconds), the model's visual quality degrades precipitously.

A primary challenge limiting the quality of autoregressive long-video generation models is a significant ***training-inference misalignment***. This misalignment manifests in two principal ways. First, a *temporal mismatch* occurs: during training, models generate short clips of up to 5 seconds—the maximum horizon of the teacher model—whereas at inference, they must generate videos of significantly greater length. Second, error accumulation caused by *supervision misalignment* during long-horizon generation. In training, the teacher model provides abundant supervision for every frame within the short clip. This intensive guidance, however, means the student model is rarely exposed to the compounding errors that naturally arise in long rollouts, leaving it ill-equipped to handle them. As a result, generation quality rapidly deteriorates beyond the 5-second training horizon, often collapsing into static or stalled content.

In this paper, we introduce **Self-Forcing++**, which directly targets the above two issues. Building upon the observation from previous works (Bruce et al., 2024; Alonso et al., 2024; Li et al., 2025a) that a teacher model, despite its own 5-second generation limit, possesses rich knowledge for correcting errors in quality-degraded videos due to its training on a vast video corpus. We leverage this insight by extending the student's generation horizon far beyond 5 seconds (up to 100 seconds in our experiments). This process intentionally produces candidate long videos that contain accumulated errors. To enable the student model to handle these errors, we then re-inject noise into these degraded rollouts and apply distribution-matching distillation with the strong teacher model, a process combined with a long-horizon rolling KV cache and windowed sampling. This strategy teaches the student to recover from degraded states and sustain high-quality, coherent video generation over extended durations. Experimental results demonstrate that our method can scale video generation up to 100 seconds, a 20× increase over the baseline, while maintaining high visual quality. By scaling up computation through extended training, our method is capable of generating videos up to 4 minutes and 15 seconds[1], utilizing 99.9% of the base model's positional embedding capacity and representing a 50× improvement over the baseline. Furthermore, our investigation revealed that the widely used VBench (Huang et al., 2024) benchmark exhibits a bias that favors over-exposed and degraded

---

[1]The maximum number of latent frames Wan2.1-T2V-1.3B supports is 1024, since we generate videos in a trunk size of 3, the maximum length we can reach is 1023 which is 99.9% of maximally supported length 1024.

frames when evaluating long videos, undermining the reliability of its results. To remedy this, we propose a new metric, Visual Stability, designed to systematically capture both quality degradation and over-exposure in long video generation. Our work paves the way for building more robust and reliable long video generation models. Our contributions are summarized as follows:

- *Identifying Horizon Scaling Bottlenecks*: We reveal the primary obstacle to extending the generation horizon of autoregressive models: a dual mismatch in temporality and supervision during training versus inference. This insight provides a clear target for overcoming previous limitations on the generation length.

- *A Simple Solution*: We propose a simple training framework, named **Self-Forcing++**. By generating beyond the teacher's horizon and correcting the student model on its own long, error-accumulated rollout trajectories, Self-Forcing++ extends high-quality video generation to 100 seconds, far surpassing previous state-of-the-art methods without reusing overlapping frames.

- *SOTA Performance and Horizon Scalability*: Self-Forcing++ achieves state-of-the-art (SOTA) performance in long-video generation across a range of durations (e.g., 10s, 50s, 100s). Furthermore, we discover a significant scaling property: by scaling the training computation, our model's generation capability extends to multiple minutes, a feat previously considered out of reach.

## 2 RELATED WORKS

**Long Video Generation** Despite their success in short video generation, DiT-based models are costly to train and infer, and most remain limited to 5–10 s outputs. Several methods aim to extend generation length (Jiang et al., 2025; Kodaira et al., 2025; Fang et al., 2025; Lin et al., 2025). RI-FLEx (Zhao et al., 2025) offers a training-free solution by revisiting positional encodings, doubling horizon length and outperforming prior approaches (Chen et al., 2023; Peng et al., 2023; Zhuo et al., 2024). Autoregressive strategies tackles it from a different perspective: Pyramid-Flow (Jin et al., 2025) links flows across pyramidal resolutions for end-to-end autoregressive generation; SkyReels-V2 (Chen et al., 2024) applies diffusion forcing for potentially infinite rollouts; MAGI-1 (Teng et al., 2025) predicts consecutive segments via progressive chunk denoising; CausVid (Yin et al., 2025) leverages block causal attention and KV caching; and Self-Forcing (Huang et al., 2025) further aligns training with inference through self-rollout during training, yielding higher quality. APT2 (Lin et al., 2025) adopts an adversarial training paradigm that first transforms a bidirectional model into a one-step autoregressive generator, then enhances generation quality through joint training with student forcing and a discriminator trained on real and generated video segments. In contrast, our method relies on pretrained diffusion models and does not require a real training dataset.

**Reinforcement Learning** Reinforcement learning is central to post-training large language models (Ouyang et al., 2022; Kumar et al., 2024; Hou et al., 2025; Xie et al., 2025) and has proven similarly effective for generative models. Early efforts focused on image-specific rewards such as ImageReward (Xu et al., 2023), Pick-a-Pic (Kirstain et al., 2023), and HPS V2 (Wu et al., 2023), later extended to video with VideoReward (Liu et al., 2025b) and VisionReward (Xu et al., 2024) for temporal and motion assessment. Optimization techniques from LLMs including DPO (Rafailov et al., 2023; Wallace et al., 2024) and GRPO (Shao et al., 2024; Liu et al., 2025a; Xue et al., 2025) have since been adapted to diffusion model.

## 3 METHODOLOGY

This section details our methodology for long video generation. We begin by revisiting the conversion of bidirectional models into streaming autoregressive generators (Yin et al., 2025; Huang et al., 2025). Building upon this, we introduce our novel strategies tailored for long-form video synthesis. The complete generative process is formalized in Algorithm 1.

### 3.1 BACKGROUND

Video diffusion models, while powerful, typically require denoising along a multi-step noise schedule, which renders the generation process computationally intensive. A prevalent strategy to mitigate this computational burden is to distill the foundational model into a few-step generator. Prominent approaches in this domain include Distribution Matching (DM) (Yin et al., 2024b;a; Luo et al.,

2025b) and Consistency Models (CM) (Song et al., 2023; Wang et al., 2024). Building upon the methodologies of CausVid and Self-Forcing, we distill the original bidirectional teacher model into a few-step generator, then convert it into an autoregressive model. This conversion is accomplished by training a student model to replicate the Ordinary Differential Equation (ODE) trajectories sampled from the teacher. We refer to this procedure as an initialization stage (see appendix A.5 for implementation details). The Self-Forcing method extends this approach by training the distilled model on self-generated rollouts of up to five seconds using techniques such as Distribution Matching Distillation (DMD) loss (Yin et al., 2024a). While this technique effectively mitigates the over-exposure artifacts present in CausVid, it exhibits a critical limitation: a significant degradation in generative quality when producing sequences that exceed its constrained training horizon.

## 3.2 Extend training beyond teacher's limit

**Motivation** As discussed earlier, the teacher model is trained exclusively on five-second video segments. Consequently, distillation-based methods such as CausVid (Yin et al., 2025) and Self-Forcing (Huang et al., 2025) only enforce student-teacher distribution alignment within this limited temporal window. This constrained training objective leads to a precipitous decline in quality when generation extends beyond this five-second horizon. Despite this performance collapse, we make a critical observation: **videos rolled out beyond the training horizon often retain structural coherence**, even if this coherence manifests as undesirable artifacts such as motion stagnation (a common failure mode in Self-Forcing). This suggests that the core problem is not a fundamental breakdown of the autoregressive mechanism, which correctly leverages the history KV cache to maintain context. Rather, the primary issue is the compounding of autoregressive errors during extended rollouts. These errors accumulate and eventually manifest as motion loss, scene freezing, and catastrophic degradation of visual fidelity. This insight motivates us to introduce a simple yet effective method to mitigate error accumulation, which is described in the following sections.

**Backwards Noise Initialization** A central challenge in extending student-teacher distillation to long-horizon video generation resides in the noise initialization strategy. In the short-horizon setting (i.e., for videos with a length up to $M$ frames), the student model can be directly supervised on complete trajectories sampled from the teacher, each originating from random noise. However, for long-horizon generation, a trajectory initialized from pure random noise is decoupled from the preceding video content, leading to a fundamental context misalignment since the sampled noise does not preserve the temporal dependencies of previously generated frames. Based on the observation mentioned above, we add noise back to the denoised latent vectors and use it as the starting noise which is also shown to boost the performance of distillation (Yin et al., 2024a). While similar techniques of re-injecting noise have been employed in prior work (Yin et al., 2024a; 2025; Huang et al., 2025), our motivation and application are distinct. Whereas they used this for short-video distillation, primarily to enhance single-shot quality or circumvent the need for real training data. We leverage it as a mechanism to enforce temporal consistency across long videos. Specifically, the student model is first rolled out to a sequence of $N$ clean frames, with $N \gg M$, where $M$ denotes the maximum horizon the teacher can reliably generate such as 5 seconds. We then re-inject noise into the student roll-out according to the same diffusion noise schedule $\{\sigma_t\}_{t=1}^T$. Formally, given the clean trajectory $\{x_i\}_{i=1}^N$ generated by the student, the generation is perturbed as:

$$x_{i,t} = (1 - \sigma_t)x_{i,0} + \sigma_t\epsilon, \quad \text{where } x_{i,0} = x_{i,t-1} - \sigma_{t-1}\,\hat{\epsilon}_\theta(x_{i,t-1}, t-1), \tag{1}$$

and $\epsilon \sim \mathcal{N}(0, I)$ denotes Gaussian noise, and $\hat{\epsilon}_\theta$ is the noise prediction network parameterized by $\theta$ which serves as the initial state for computing the teacher and student distributions. This approach ensures that the distribution divergence between student and teacher model is evaluated on trajectories that retain temporal consistency and correctly structured.

**Extended Distribution Matching Distillation** Our strategy for extending training to long videos is grounded in the observation that although the bidirectional teacher model is trained exclusively on short, five-second clips, it implicitly captures the underlying data distribution of the "world" from its training data. From this perspective, any short, contiguous video segment can be viewed as a sample from the marginal distribution of a valid, longer video sequence (Bruce et al., 2024; Alonso et al., 2024; Li et al., 2025a). This intuition motivates our core methodological extension. Since our baseline method Self-Forcing (Huang et al., 2025) restricts the training duration to the first $M$ frames (typically ~5 seconds), we instruct the student model to roll out to $N$ frames where $N \gg M$. We

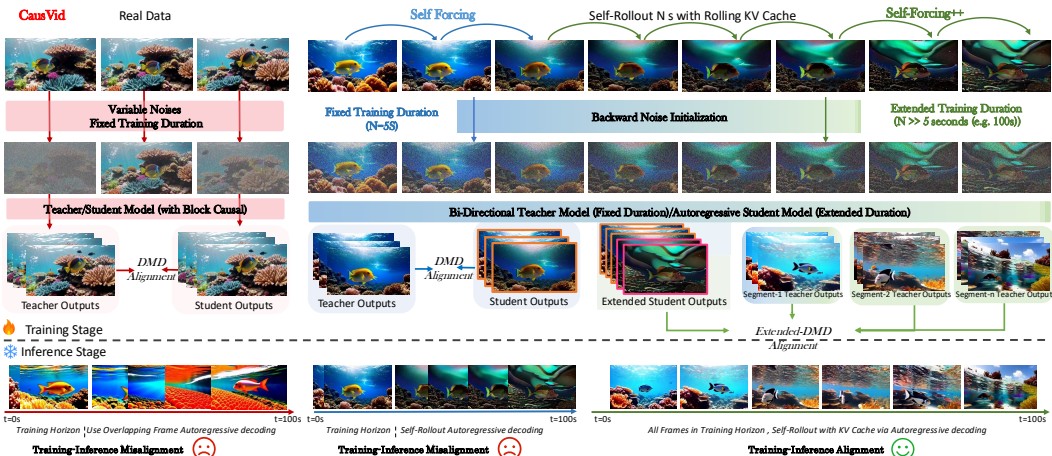

Figure 2: Workflow between baselines and Self-Forcing++. Our method employ backward noise initialization, extended DMD and rolling KV Cache to effectively mitigates train-test discrepancies.

then uniformly sample a contiguous window of length $T$ from the generated sequence, and compute the distributional discrepancy between the student and teacher models within this window. This sliding-window distillation process is formalized as eq. (2):

$$
\nabla_\theta \mathcal{L}_{\underset{\text{extended}}{\text{DMD}}} = \mathbb{E}_t \mathbb{E}_z \left[ \nabla_\theta \, \mathrm{KL}\Big( p^S_{\theta,t}(z) \,\|\, p^T_t(z) \Big) \right]
$$

$$
\approx - \mathbb{E}_t \, \mathbb{E}_{i \sim \mathrm{Unif}\{1,\dots,N-K+1\}} \left[ \int \Big( s^T\big(\Phi(G_\theta(z_i),t),t\big) - s^S\big(\Phi(G_\theta(z_i),t),t\big) \Big) \frac{dG_\theta(z_i)}{d\theta} \, dz_i \right],
\tag{2}
$$

Here, $G_\theta(z)$ denotes the student generator rollout given a latent $z$, and $\Phi(\cdot, t)$ is the transformation process at timestep $t$. $p^S_{\theta,t}$ and $p^T_t$ represent the student and teacher distributions at time $t$, with corresponding scores $s^S$ and $s^T$. We uniformly sample a starting index $i \sim \mathrm{Unif}\{0,\dots,N-K\}$ from the student rollout of length $N$, and extract a window of length $K$. The student is then trained to minimize the average KL divergence between its distribution and the teacher's distribution across this window. The window size $K$ is typically chosen to match the horizon to which the teacher model was originally trained to generate.

***Remark*** *Bi-directional diffusion can be seen as a process to gradually restore a degraded target in different denoising **time-steps**. Our method adapts the idea to autoregressive video generation regime by having a short-horizon teacher gradually restore student's degraded rollouts at different temporal **time-frames** and then distills these correction knowledge back into the student model.*

**Training with rolling KV Cache**  Despite using KV cache at inference time, CausVid (Yin et al., 2025) still relies on recomputing overlapping frames and suffers from a severe over-exposure problem. Self-Forcing (Huang et al., 2025) attempts to address this but introduces a train-inference mismatch by using a fixed cache during training and a rolling cache at inference. Although this is partially mitigated by masking the first latent frame, the mismatch still leads to substantial error accumulation and temporal flickering in long videos (see fig. 4). In contrast, our method naturally eliminates this mismatch by employing a rolling KV cache during both training and inference. At training time, this cache is used to roll out sequences far beyond the teacher's supervisory horizon to compute the extended DMD as detailed above. Consequently, our approach greatly simplifies the entire process, requiring neither the recomputation of overlapping frames nor latent frame masking.

### 3.3 IMPROVING LONG-TERM SMOOTHNESS VIA GRPO

A common drawback of generative models (Xiao et al., 2023; Luo et al., 2025a) employing sliding-window or sparse attention mechanisms for long sequences generation is the gradual loss of long-term memory. This degradation often manifests as temporal inconsistencies such as objects abruptly

emerging or vanishing or unnaturally rapid scene transitions. Although the method we proposed above has achieved strong results, we show that Group Relative Policy Optimization (GRPO), a reinforcement learning technique (Liu et al., 2025a; Xue et al., 2025), can be utilized in autoregressive video generation framework when such phenomenon presents. The per step importance weight $\rho_{t,i} = \frac{\pi_\theta(a_{t,i} \mid s_{t,i})}{\pi_{\theta_{\text{old}}}(a_{t,i} \mid s_{t,i})}$ where $\pi_\theta(a_{t,i}|s_{t,i})$ denotes the policy function for output $o_i$ at time step $t$ can be computed according to eq. (1) and the overall generation probability can be computed as the sum of all the log probabilities in current autoregressive rollouts which we show in appendix A.6. To guide the optimization process towards temporally smooth outputs, we follow prior work (Chefer et al., 2025; Nam et al., 2025) and use the relative magnitude of optical flow between consecutive frames as a proxy for motion continuity.

---

**Algorithm 1** Self-Forcing++ with Backward Noise Initialization (ours)

---

**Require:** Student $G_\theta$, teacher $T_\phi$, cache size $L$; rollout length $N \gg 5s$; slice length $K$ (5s); denoise steps $\{t_1, \ldots, t_T\}$
1: **loop**
2:      $V \leftarrow \text{Rollout}(G_\theta, N, L)$
3:      Pick $i \sim \{1, \ldots, N{-}K{+}1\}$, set $W \leftarrow V[i : i{+}K{-}1]$                ▷ uniform slice
4:      Sample $t \sim \{t_1, \ldots, t_T\}$
5:      Backward noise initialization: $x_t(W) \leftarrow \text{BackwardNoiseInit}(W, t)$
6:      $\mathcal{L}_{\text{DMD}} \leftarrow \text{DMD}(G_\theta(x_t(W), t), \, T_\phi(x_t(W), t))$
7:      $\theta \leftarrow \theta - \eta \nabla_\theta \mathcal{L}_{\text{DMD}}$
8: **end loop**
9: $R \leftarrow \text{OpticalFlowReward}(G_\theta); \quad \theta \leftarrow \text{GRPO\_update}(\theta, R)$

---

### 3.4 NEW METRICS FOR LONG VIDEOS EVALUATION

Most prior works rely on VBench (Huang et al., 2024) to assess image and aesthetic quality in long video generation. We find, however, that outdated evaluation models make the benchmark favor over-exposed videos (e.g., CausVid) and degraded long videos (e.g., Self-Forcing), leading to inaccurate scores. To address this, we adopt Gemini-2.5-Pro (Comanici et al., 2025), a state-of-the-art video MLLM with strong reasoning ability (Chiang et al., 2024; Liu et al., 2025c). Our protocol defines key long-video issues such as over-exposure and error accumulation, prompts Gemini-2.5-Pro to rate videos along these axes, and aggregates the results onto a 0–100 scale termed visual stability for consistent comparison. More details are provided in fig. 3 and appendix A.8.

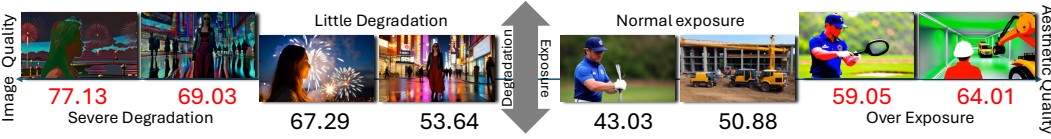

Figure 3: The left figure shows the issue of image score issue and the right figure shows the issue of aesthetic score of regular and degraded images from earlier and later frames of the same video. VBench tends to overrate degraded and over-exposed frames rendering these two metrics unreliable.

## 4 EXPERIMENTAL RESULTS

### 4.1 SETTINGS

**Baseline methods** We include the following baseline methods such as NOVA (Deng et al., 2025), Pyramid Flow (Jin et al., 2025), SkyReels-V2-1.3B (Chen et al., 2025), MAGI-1-4.5B (Sand-AI, 2025) distilled to 16 steps for long video generation, CausVid (Yin et al., 2025) and Self-Forcing (Huang et al., 2025), both 1.3B distilled few-step generators similar to ours. Additional two state-of-the-art bidirectional models LTX-Video (HaCohen et al., 2024) and Wan2.1 (Wan et al., 2025) are included for references.

**Evaluation metrics** We conduct evaluations under two primary settings. The first setting follows the general VBench protocol (Huang et al., 2024), which measures generation quality on short videos

of 5 seconds using 946 prompts across 16 dimensions. The second setting examines the model's capacity to extend generation up to 50/75/100 seconds with the same prompt set used in CausVid, consisting of 128 prompts from MovieGen (Polyak et al., 2024). Performance in this setting is assessed with both VBench Long and our proposed improved evaluation metric.

## 4.2 EMPIRICAL RESULTS IN LONG VIDEO GENERATION

Quantitative and qualitative results are presented in tables 1 and 2, and figs. 1 and 4, respectively. Our method achieves competitive performance in short-horizon generation and demonstrates substantial advantages as the generation horizon extends.

**Short-Horizon (5s)**: Although not specifically trained for the initial 5 seconds, our model performs comparably to Self-Forcing on short clips, achieving strong overall results with a semantic score of 80.37 and a total score of 83.11, both surpassing the remaining baselines.

Table 1: Performance comparisons on 5s short videos and 50s long videos. Baseline methods achieve high temporal quality scores primarily due to stagnation reflected by their dynamic degree.

| Model | #Params | Throughput (FPS) ↑ | Results on 5s ↑ | | | Results on 50s ↑ | | | | |
|---|---|---|---|---|---|---|---|---|---|---|
| | | | Total Score | Quality Score | Semantic Score | Text Alignment | Temporal Quality | Dynamic Degree | Visual Stability | Framewise[†] Quality |
| *Bidirectional models* | | | | | | | | | | |
| LTX-Video | 1.9B | 8.98 | 80.00 | 82.30 | 70.79 | - | - | - | - | - |
| Wan2.1 | 1.3B | 0.78 | 84.67 | 85.69 | 80.60 | - | - | - | - | - |
| *Autoregressive models* | | | | | | | | | | |
| NOVA | 0.6B | 0.88 | 80.12 | 80.39 | 79.05 | 24.58 | 86.53 | 31.96 | 45.94 | 34.45 |
| Pyramid Flow | 2B | 6.7 | 81.72 | **84.74** | 69.62 | - | - | - | - | - |
| MAGI-1 | 4.5B | 0.19 | 79.18 | 82.04 | 67.74 | 26.04 | 88.34 | 28.49 | 51.25 | 54.20 |
| SkyReels-V2 | 1.3B | 0.49 | 82.67 | 84.70 | 74.53 | 23.73 | 88.78 | 39.15 | 60.41 | 54.13 |
| CausVid | 1.3B | 17.0 | 82.46 | 83.61 | 77.84 | 25.25 | 89.34 | 37.35 | 40.47 | 61.56 |
| Self Forcing | 1.3B | 17.0 | 83.00 | 83.71 | 80.14 | 24.77 | 88.17 | 34.35 | 40.12 | 61.06 |
| Ours | 1.3B | 17.0 | **83.11** | 83.79 | **80.37** | **26.37** | **91.03** | **55.36** | **90.94** | 60.82 |

- indicates that the model either fails to generate videos at the specified length or that the output collapses into random noise.
† As discussed in section 3.4, framewise quality is unreliable for long videos, we include it here for reference.

**Long-Horizon (50s/75s/100s)**: The superiority of our method becomes more pronounced in long-horizon generation. We observe consistent improvements across key metrics. E.g. our model achieves a text alignment score of 26.04 and dynamic degree of 54.12 with 100-second video, outperforming CasuVid by 6.67% and 56.4% respectively which relies on recomputing overlapping frames and our baseline method Self-Forcing by 18.36% and 104.9% respectively as shown in fig. 4. This suggests that our approach effectively mitigates error accumulation during long rollouts.

Table 2: Performance comparisons on 75s and 100s long videos. Baseline methods achieve high temporal quality scores primarily due to stagnation or degrade to pure noise.

| Model | Results on 75s ↑ | | | | | Results on 100s ↑ | | | | |
|---|---|---|---|---|---|---|---|---|---|---|
| | Text Alignment | Temporal Quality | Dynamic Degree | Visual Stability | Framewise Quality | Text Alignment | Temporal Quality | Dynamic Degree | Visual Stability | Framewise Quality |
| *Autoregressive models* | | | | | | | | | | |
| NOVA | 23.37 | 86.32 | 31.24 | 34.06 | 31.53 | 22.89 | 86.24 | 31.09 | 32.97 | 31.03 |
| MAGI-1 | 24.95 | 87.89 | 24.82 | 43.28 | 52.04 | 23.75 | 87.62 | 22.21 | 39.38 | 50.90 |
| SkyReels-V2 | 22.70 | 88.99 | 39.89 | 55.47 | 51.55 | 22.05 | 88.80 | 38.75 | 56.72 | 50.48 |
| CausVid | 24.76 | 89.14 | 35.82 | 39.84 | 60.96 | 24.41 | 89.06 | 34.60 | 39.21 | 61.01 |
| Self Forcing | 23.39 | 87.79 | 29.15 | 35.00 | 60.02 | 22.00 | 87.39 | 26.41 | 32.03 | 58.25 |
| Ours | **26.31** | **91.00** | **55.62** | **86.10** | 60.67 | **26.04** | **90.87** | **54.12** | **84.22** | 60.66 |

In contrast, baseline methods exhibit significant degradation when generating long videos. Their primary failure modes are: **i)** Motion Collapse: While maintaining short-term temporal structure, their videos frequently collapse into nearly static sequences, as reflected by their low dynamic degree scores. Our method, however, sustains coherent motion throughout the entire sequence. **ii)** Fidelity Degradation: Baselines often suffer from exposure instability. For instance, CausVid trends towards over-exposure, while Self-Forcing videos progressively darken. Our model maintains stable brightness and visual quality. This degradation in Self-Forcing is a direct consequence of accumulated

errors without explicit long-horizon training. While some diffusion forcing methods show sporadic recovery from noise collapse such as SkyReels, the resulting content is of low fidelity.

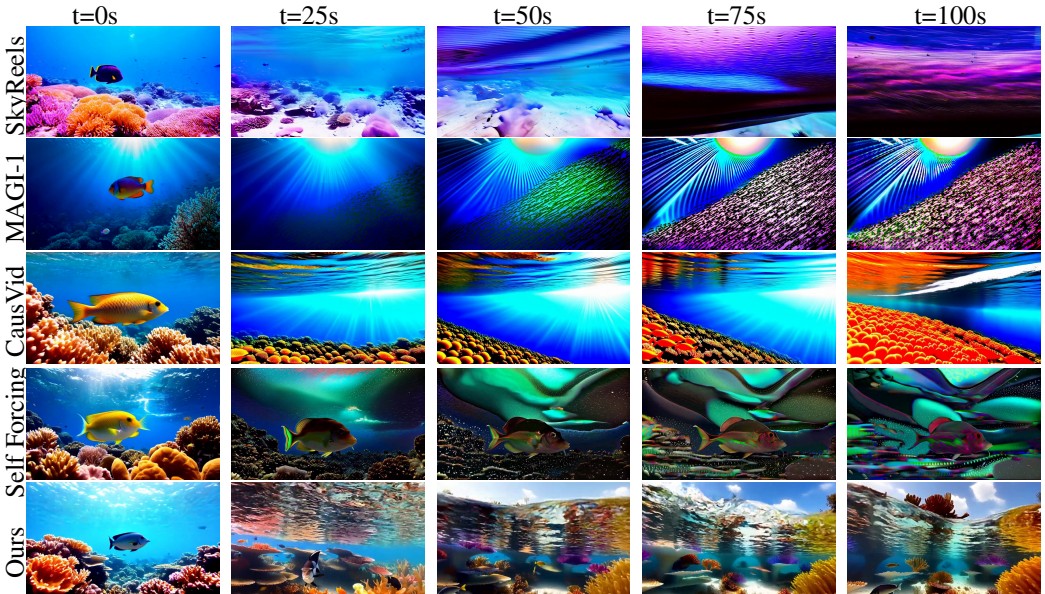

Figure 4: 100-second video generated for prompt "A vibrant tropical fish glides gracefully through colorful ocean reefs, surrounded by swaying coral...". Baseline methods usually suffer from error accumulation and over-exposure, causing severe quality degradation when generating long videos.

## 4.3 ABLATION STUDY

### 4.3.1 LENGTH OF ATTENTION WINDOW

A straightforward way to mitigate Self-Forcing's training–inference mismatch is to shorten the attention span during training, exposing the model to more diverse cache states within a limited horizon. For instance, a 5-second clip corresponds to 21 latent frames; by reducing the attention window, the model is forced to slide attention multiple times. As shown in table 3 and visualized in Appendix fig. 7, smaller windows bring modest gains. For example, visual stability improves from 40.12 to 52.50 with a window of 9 latent frames. However, this comes at the cost of increased inconsistency, since the model now relies on much less context compared to the original 21-frame history.

Table 3: Ablation study on various methods to reduce error accumulation measured by visual stability on 50s videos.

| Causvid | Self-Forcing | Attn-15 | Attn-12 | Attn-9 | Ours |
|---------|--------------|---------|---------|--------|------|
| 40.47 | 40.12 | 44.69 | 42.19 | 52.50 | **90.94** |

### 4.3.2 THE IMPACT OF GRADIENT STEPS

Our method builds upon the same underlying backbone as Self-Forcing (Huang et al., 2025) which is a four-step distilled generation model. In the main experiments, we enable gradients exclusively at the final denoising step. To further understand the impact of enabling gradient at different steps, we compare this setting with an alternative strategy that uniformly samples the steps at which gradients are enabled similar to that of Self-Forcing (Huang et al., 2025). We evaluate the models on 50-second generation and the results are shown in table 4.

We observe that enabling gradients over uniformly sampled steps improves visual stability and yields higher framewise quality. However, this advantage comes with a reduction in dynamism, as the resulting videos exhibit noticeably slower motion. This is likely due to that

Table 4: Ablation study on gradient step sampling.

| | Text Alignment | Temporal Quality | Dynamic Degree | Visual Stability | Framewise Quality |
|---|---|---|---|---|---|
| Uniform | 26.14 | 88.69 | 35.92 | 91.25 | 62.48 |
| Ours | 26.37 | 91.03 | 55.36 | 90.94 | 60.82 |

restricting gradient computation to only the final step provides the student model with more flexibility, allowing greater freedom. In contrast, enabling gradients at earlier steps may impose stronger alignment with the teacher model, inadvertently constraining the motion dynamics.

### 4.3.3 THE IMPACT OF WINDOW SAMPLING DISTRIBUTION

For our main method, we uniformly sample from the roll-out sequences. Here we also evaluate an alternative sampling strategy based on a $\text{Beta}$ distribution. Specifically, we draw a value $u \sim \text{Beta}(1,2)$ and map it to the target range using the transformation $L = \text{round}(\ell + u \cdot (h - \ell))$, where $\ell$ and $h$ denote the lower and upper bounds of the roll-out interval. This sampling scheme places more weight toward earlier steps in the sequence. Results comparing uniform sampling with $\text{Beta}(1,2)$ sampling on 50-second generation are provided in table 5.

It can be seen that when focusing on shorter horizons, the model still achieves a substantial improvement over baseline methods. However, it produces videos with noticeably reduced motion, e.g, a dynamic degree of 45.66 compared to 55.36 under uniform sampling. Although this setting yields a higher alignment score, it's more prone to stagnation, which in turn leads to greater error accumulation over long sequences.

Table 5: Ablation study on window sampling.

|  | Text Alignment | Temporal Quality | Dynamic Degree | Visual Stability | Framewise Quality |
|---|---|---|---|---|---|
| Self Forcing | 24.77 | 88.17 | 34.35 | 40.12 | 61.06 |
| Beta Sampling | 26.65 | 90.14 | 45.66 | 86.25 | 60.85 |
| Ours | 26.37 | 91.03 | 55.36 | 90.94 | 60.82 |

### 4.3.4 THE IMPACT OF K/N RATIO

A central limitation of Self-Forcing (Huang et al., 2025) is that its training horizon is restricted to 5 seconds, matching the teacher's supervision window. To better understand how extending this horizon affects downstream performance, we conduct an experiment that doubles the training duration to 10 seconds which exceeds the teacher's horizon and forces rolling KV while avoiding prohibitive training cost. We evaluate the model on 50-second generation and show results in table 6.

The darkening and temporal-flickering artifacts observed in fig. 4 are substantially reduced in the early portions of the sequences, consistent with the improved text-alignment score. However, because the training horizon remains short, the model is still not exposed to sufficiently diverse failure modes for the teacher to correct. As a result, the generations gradually stagnate toward near-static frames, leading to strong error accumulation over longer roll-outs.

Table 6: Ablation study on training horizon.

|  | Text Alignment | Temporal Quality | Dynamic Degree | Visual Stability | Framewise Quality |
|---|---|---|---|---|---|
| Self Forcing | 24.77 | 88.17 | 34.35 | 40.12 | 61.06 |
| 10s Horizon | 25.36 | 88.78 | 35.91 | 50.78 | 59.50 |
| Ours | 26.37 | 91.03 | 55.36 | 90.94 | 60.82 |

### 4.3.5 THE EFFECT OF GRPO WITH OPTICAL-FLOW REWARD

Here we show its effectiveness for enhancing temporal consistency by examining the optical flow magnitude, a proxy for temporal stability. As visualized in fig. 5, videos generated without GRPO may suffer from abrupt scene transitions. These transitions manifest as sharp spikes in the optical flow magnitude, an artifact that is exacerbated by the rolling window mechanism used during inference. By promoting smoother temporal transitions, our GRPO method effectively suppresses these spikes. This results in a marked improvement in long-range consistency and overall perceptual quality of the generated videos.

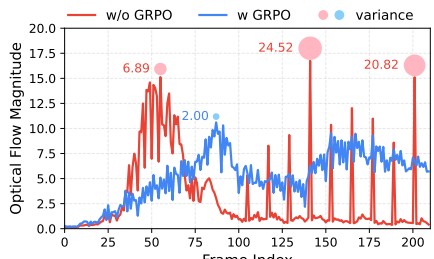

Figure 5: Comparison of video generation outcomes with and without GRPO. Variance is computed with a window size of 8 frames.

### 4.4 TRAINING BUDGET SCALING

Finally, we investigate the effect of scaling the training budget on the model's long-duration video generation capabilities. We observe that increasing either the roll-out duration or the total training

iterations improves long-term generation quality. Here, we mainly scale up the number of training iterations with a 50-second training horizon, using a batch size of 8 on 8 H100 GPUs and the same hyperparameters described in appendix A.2. As illustrated in fig. 6, our model, following ODE initialization, exhibits only a nascent ability to generate short, low-fidelity clips. We establish a baseline ($1\times$ budget) as the training required to produce a coherent 5-second video which is around 500 iterations. At this scale, extending generation leads to significant temporal flickering and error accumulation, a failure mode similar to that of Self-Forcing (Huang et al., 2025). Increasing the budget to $4\times$ enables the model to maintain semantic coherence over longer horizons, successfully rendering a consistent subject like the specified elephant. At $8\times$, the model begins to generate detailed backgrounds and more semantically accurate subjects, although motion dynamics remain limited and temporal quality degradation persists. A further scaling to $20\times$ yields a substantial improvement, producing high-fidelity videos that remain stable for over 50 seconds. Remarkably, at a $25\times$ budget with around 12500 iteration, the model successfully generates a 255-second video with negligible quality loss. These findings indicate that scaling the training budget is a viable path toward high-quality, long-duration video synthesis, circumventing the reliance on large-scale real video datasets, which are notoriously difficult to acquire.

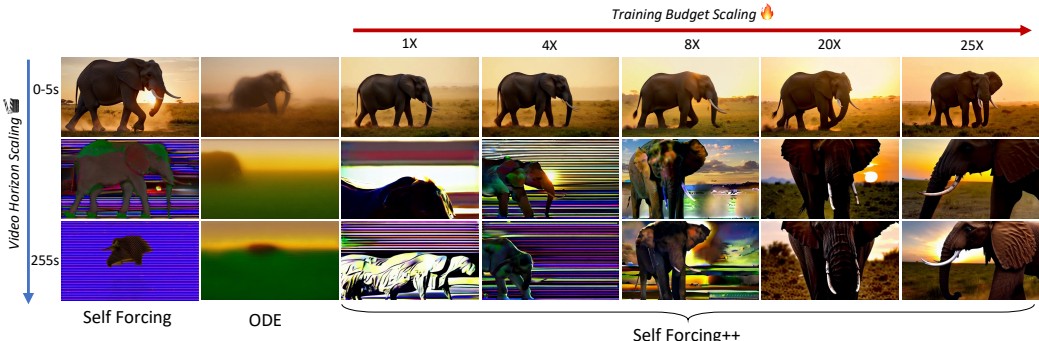

Figure 6: Scaling phenomenon observed in 255-second generation for prompt: "A massive elephant walks slowly across a sunlit savannah, dust rising around its feet, the warm glow of sunset...".

## 5 CONCLUSION

We introduce Self-Forcing++, a method that mitigates error accumulation in autoregressive long-video generation. By leveraging a short-video teacher to guide the student on its own self-generated long rollouts, our approach learns to correct errors without requiring long-video supervision. Experiments demonstrate that our method significantly extends video length to even over 4 minutes (a $50\times$ improvement over the baseline) while maintaining high fidelity. We also propose a new metric, Visual Stability, to address critical biases in existing long-video evaluation benchmarks. Our contributions pave the way for more robust and scalable long-video synthesis.

## 6 LIMITATIONS AND FUTURE WORK

Our method, while effective, inherits certain limitations from its Self-Forcing foundation and the capacity of the underlying Wan2.1-T2V-1.3B model. Key drawbacks include slower training speed compared to teacher-forcing and a lack of long-term memory, which can cause content divergence in regions occluded for extended periods. To address these challenges, we identify several promising future directions. First, to tackle the high training cost of self-rollout, we will explore parallelizing the training process. Second, to further mitigate quality degradation over long sequences, we plan to investigate techniques for controlling the fidelity of latent vectors. This includes quantizing latent representations stored in the KV cache, as suggested by prior works (Zhang & Agrawala, 2025), or normalizing the KV cache to prevent distributional shift. Finally, we aim to incorporate long-term memory mechanisms (Li et al., 2025b; Liu et al., 2025d) into our autoregressive framework, which we believe is crucial for achieving true long-range temporal coherence.

## ACKNOWLEDGMENTS

We sincerely thank the conference reviewers for their constructive feedback and helpful suggestions, and we also thank the authors of the works on which our paper builds.

## ETHICS STATEMENT

Our work builds on the open-source Wan2.1-T2V-1.3B model, which is publicly available and does not involve new dataset collection or release. No human subjects or private data are used, and we emphasize the importance of responsible use of generative models.

## REPRODUCIBILITY STATEMENT

Due to the simple design, our proposed method can be reproduced by following the described steps in algorithm 1. Our training hyperparameter are also released in the appendix for reproducibility.

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

# A APPENDIX

## A.1 THE USE OF LARGE LANGUAGE MODELS (LLMs)

Large Language Models (LLMs) such as GPT were used solely for language polishing and clarity improvements in the writing of this paper. All technical content, dataset design, experimental results, and analyses were created by the authors. The models were not used to generate ideas, data, or experimental outcomes.

## A.2 DETAILED EVALUATION RESULTS FOR ALL DIMENSIONS

Due to limited space, we only report VBench aggregated metrics in tables 1 and 2 in the main text. Here we show the full evaluated data for each dimension in table 7.

Table 7: Comparison of models across multiple quality metrics on 50s, 75s, and 100s videos for our main results. The gray metrics are aggregated metrics by VBench Long.

| | text alignment | overall consistency | clip score | temporal quality | subject consistency | background consistency | motion smoothness | dynamic degree | frame-wise quality | aesthetic quality | imaging quality |
|---|---|---|---|---|---|---|---|---|---|---|---|
| *50 seconds* | | | | | | | | | | | |
| NOVA | 24.58 | 20.55 | 28.61 | 86.53 | 92.48 | 95.21 | 99.18 | 31.96 | 34.45 | 39.85 | 29.06 |
| MAGI-1 | 26.04 | 21.55 | 30.53 | 88.34 | 98.08 | 97.50 | 99.36 | 28.49 | 54.20 | 52.15 | 56.26 |
| SkyReels-V2 | 23.73 | 18.94 | 28.53 | 88.78 | 96.06 | 96.43 | 98.67 | 39.15 | 54.13 | 50.44 | 57.82 |
| CausVid | 25.25 | 20.71 | 29.80 | 89.34 | 98.29 | 97.27 | 98.45 | 37.35 | 61.56 | 57.91 | 65.21 |
| Self-Forcing | 24.77 | 20.27 | 29.27 | 88.17 | 96.77 | 96.24 | 98.40 | 34.35 | 61.06 | 54.95 | 67.17 |
| Ours | 26.37 | 22.03 | 30.71 | 91.03 | 97.00 | 95.55 | 98.39 | 55.36 | 60.82 | 53.76 | 67.87 |
| *75 seconds* | | | | | | | | | | | |
| NOVA | 23.37 | 19.19 | 27.54 | 86.32 | 91.78 | 95.57 | 99.14 | 31.24 | 31.53 | 37.34 | 25.73 |
| MAGI-1 | 24.95 | 20.36 | 29.53 | 87.89 | 98.20 | 97.70 | 99.29 | 24.82 | 52.04 | 49.38 | 54.71 |
| SkyReels-V2 | 22.70 | 17.60 | 27.81 | 88.99 | 96.08 | 96.54 | 98.88 | 39.89 | 51.55 | 47.55 | 55.55 |
| CausVid | 24.76 | 19.99 | 29.53 | 89.14 | 98.32 | 97.28 | 98.50 | 35.82 | 60.96 | 56.87 | 65.06 |
| Self-Forcing | 23.39 | 18.85 | 27.93 | 87.79 | 97.43 | 96.50 | 98.77 | 29.15 | 60.02 | 53.28 | 66.77 |
| Ours | 26.31 | 22.09 | 30.53 | 91.00 | 96.93 | 95.45 | 98.29 | 55.62 | 60.67 | 53.38 | 67.97 |
| *100 seconds* | | | | | | | | | | | |
| NOVA | 22.89 | 18.63 | 27.15 | 86.24 | 91.66 | 95.50 | 99.13 | 31.09 | 31.03 | 36.64 | 25.42 |
| MAGI-1 | 23.75 | 18.94 | 28.57 | 87.62 | 98.35 | 97.99 | 99.20 | 22.21 | 50.90 | 47.25 | 54.55 |
| SkyReels-V2 | 22.05 | 16.84 | 27.25 | 88.80 | 96.05 | 96.52 | 98.86 | 38.75 | 50.48 | 46.33 | 54.62 |
| CausVid | 24.41 | 19.61 | 29.22 | 89.06 | 98.41 | 97.46 | 98.54 | 34.60 | 61.01 | 57.22 | 64.79 |
| Self-Forcing | 22.00 | 17.16 | 26.84 | 87.39 | 97.39 | 96.76 | 98.52 | 26.41 | 58.25 | 51.16 | 65.35 |
| Ours | 26.04 | 21.75 | 30.34 | 90.87 | 97.09 | 95.53 | 98.35 | 54.12 | 60.66 | 53.00 | 68.31 |

### IMPLEMENTATION DETAILS

We adopt the same base model Wan2.1-T2V-1.3B (Wan et al., 2025) as Causvid and Self-Forcing, which is later converted into an autoregressive model as describe above. The model is initialized with sampled 16K ODE training trajectories by optimizing the loss in eq. (4). We use the same filtered and LLM-extended version of VidProM (Wang & Yang, 2024) as Self-Forcing for training. In the training phase, since we utilize backward noise initialization, thus we don't need real data for training. We utilize the same Wan2.1-T2V-1.3B as the teacher model.

### TRAINING DETAILS

We use 8 H100 GPUs each with 80GB GPU memory during training with a training batch size of 8 and training length 100 seconds for a total of 48 H100 GPU days for full training. The cost could be further reduced when training with LoRA (Hu et al., 2022). For inference, it takes around 50 seconds to generate a 50-second video which occupies around 22GB of memory on one H100 GPU with 80GB memory. The hyperparameters are mostly adopted from Self-Forcing such as the denoising steps of 1000,750,500 and 250 with a generator learning rate of $2e^{-6}$ and critic learning rate of $4e^{-7}$. The generator and critic update ratio is 5. AdamW optimizer is used for both generator and critic both with $\beta_1 = 0$ and $\beta_2 = 0.999$. Our rolling KV cache window size is 21 latent frames in all cases except ablation study. The model is updated with EMA starting at 200 epochs. We have also inspected the version without EMA which can also generate long high quality videos but the EMA version performs better. Our method can already generate consistent high quality long videos such as videos up to 4minute 15 seconds before GPRO, in the ablation study, we show that it's possible to further boost the model's performance with properly designed rewards.

### A.3 Adding noise to context window

As demonstrated in table 1, methods such as MAGI-1 (Sand-AI, 2025) and SkyReels-V2 (Chen et al., 2025), which rely on variable noisy context injection following a predefined schedule (Chen et al., 2024), are insufficient to mitigate error accumulation when rolling to long videos. To further investigate its effect on methods, we conduct an experiment where noise is manually injected into the KV cache to explicitly simulate the effect of accumulated errors over extended sequences. Specifically, before adding any query or key to the KV cache, we inject random Gaussian noise into them. While this strategy yields a slight improvement in both image quality and visual stability compare to the original Self-Forcing, it nonetheless fails to prevent substantial degradation in long-horizon video generation which can be seen in fig. 7.

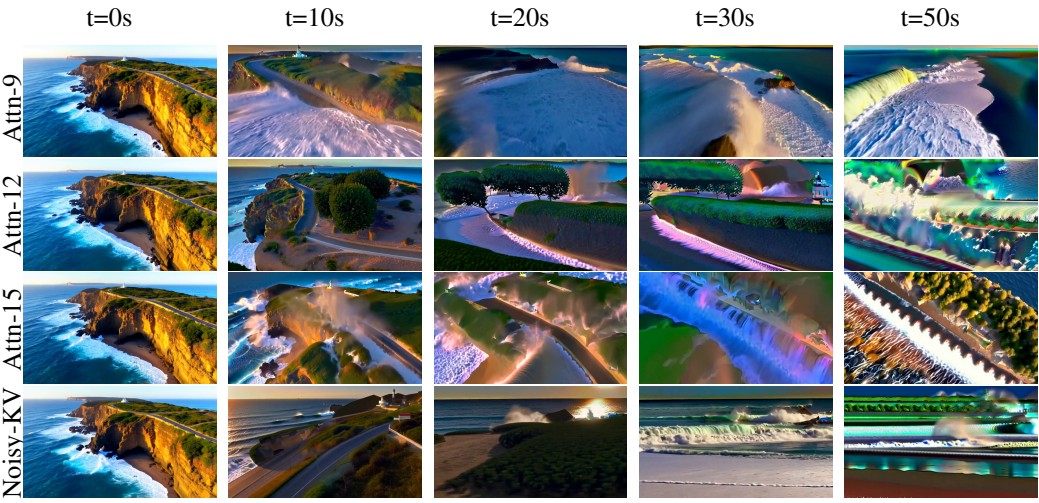

Figure 7: Ablation study for various methods of mitigating error accumulation. Here is one visualization of generated 50-second video for prompt: "Drone view of waves crashing against the rugged cliffs along Big Sur's garay point beach..."

### A.4 More Related Works

**Video Diffusion Models** Video diffusion models have advanced rapidly, beginning with UNet (Ronneberger et al., 2015) based approaches that extended image diffusion backbones into the temporal domain (Ho et al., 2022a;b; He et al., 2022; Blattmann et al., 2023). These early models enabled short-form video generation but faced limitations in scalability. The introduction of the Diffusion Transformer (DiT)(Peebles & Xie, 2023) represented a turning point by replacing convolutional hierarchies with transformer blocks, which allowed models to capture global spatio-temporal dependencies more effectively and to scale with larger datasets and computational resources. This shift led to a new wave of architectures, such as Sora(OpenAI, 2024), which produces realistic, coherent videos with strong temporal consistency and diverse motion, and Hunyuan Video (Kong et al., 2024), which employs a causal 3D VAE (Kingma & Welling, 2013) for spatio-temporal token compression in latent space combined with a large language model for text conditioning. Wan 2.1 (Wan et al., 2025) further demonstrates the benefits of massive pretraining for high-resolution video generation. CogVideoX (Hong et al., 2022; Yang et al., 2024) introduces an expert transformer with adaptive LayerNorm to enhance cross-modal fusion, supported by a 3D VAE, progressive training, and multi-resolution frame packing, thereby achieving strong text alignment and motion coherence. Open-Sora (Lin et al., 2024; Peng et al., 2025) and Open-Sora-Plan (Lin et al., 2024) extend these advancements in the open-source community, delivering high-quality video generation and significantly accelerating progress in efficiency and realism. Collectively, these works illustrate how scaling strategies and architectural innovations have transformed video diffusion from modest UNet adaptations into transformer-driven models capable of generating controllable and high-quality videos.

**Long Video Generation** Due to the substantial training and inference cost of DiT-based architectures, most state-of-the-art models remain limited to generating videos of 5–10 seconds. To overcome this constraint, a number of techniques have been introduced to extend generation to longer durations (Jiang et al., 2025; Kodaira et al., 2025; Fang et al., 2025; Lin et al., 2025). RIFLEx (Zhao et al., 2025) is a training-free approach that revisits positional encoding, effectively doubling the generation length by avoiding encodings that induce repetitive motion, and surpassing prior methods by a large margin (Chen et al., 2023; Peng et al., 2023; Zhuo et al., 2024). Another promising direction is autoregressive video generation. Nova (Deng et al., 2024) reformulates video synthesis as a non-quantized autoregressive problem, jointly modeling temporal frame-by-frame prediction and spatial set-by-set prediction, which enables flexible in-context learning. Pyramid-Flow (Jin et al., 2025) interprets denoising as a hierarchical process across multi-stage pyramids, linking flows across resolutions and time to support end-to-end autoregressive video generation with a single diffusion transformer. SkyReels-V2 adopts diffusion forcing (Chen et al., 2024) to support potentially infinite rollouts, while MAGI-1 (Teng et al., 2025) trains a model to progressively denoise per-chunk noise that increases over time, autoregressively predicting fixed-length segments of consecutive frames. CausVid (Yin et al., 2025) employs block causal attention and a KV cache to autoregressively extend sequences, and Self-Forcing (Huang et al., 2025) further aligns training with inference by incorporating the KV cache directly during training, producing high-quality short videos.

**Reinforcement Learning** Reinforcement learning has become a central component in the post-training of large language models (Ouyang et al., 2022; Kumar et al., 2024; Hou et al., 2025; Xie et al., 2025). With the rise of image generation, it has also proven effective for improving generative models, with early efforts introducing reward models tailored to images, such as ImageReward (Xu et al., 2023), Pick-a-Pic (Kirstain et al., 2023), and HPS V2 (Wu et al., 2023). These concepts have since been extended to video through reward functions like VideoReward (Liu et al., 2025b) and VisionReward (Xu et al., 2024), which assess temporal coherence and motion quality. Building on these reward signals, optimization techniques first developed for language models, including Direct Preference Optimization (DPO)(Rafailov et al., 2023) and Group Relative Policy Optimization (GRPO)(Shao et al., 2024), have been adapted to diffusion-based generation. Notably, Diffusion-DPO (Wallace et al., 2024) applies preference-based training directly to diffusion models, while Flow-GRPO (Liu et al., 2025a; Xue et al., 2025) leverages GRPO to fine-tune video diffusion models, resulting in improvements in both visual fidelity and motion consistency.

## A.5 More Background

Video diffusion models are typically trained to denoise along a fixed noise schedule, which makes video generation computationally expensive. A common strategy to reduce this cost is to distill the model into a few-step generator, using approaches such as Distribution Matching (Yin et al., 2024b;a; Luo et al., 2025b) and Consistency Model (Song et al., 2023; Wang et al., 2024). In line with CausVid and Self-Forcing, we also adopt DMD to distill the original bidirectional model into a few-step model, which can be viewed as minimizing the reverse KL divergence between the student and teacher models, as formulated in eq. (3).

$$
\begin{aligned}
\nabla_\theta \mathcal{L}_{\text{DMD}} &= \mathbb{E}_t \Big[ \nabla_\theta \, \text{KL}(p_{\text{fake},t} \, \| \, p_{\text{real},t}) \Big] \\
&= -\mathbb{E}_t \bigg[ \int \Big( s_{\text{real}}(\Phi(G_\theta(z),t),t) - s_{\text{fake}}(\Phi(G_\theta(z),t),t) \Big) \frac{dG_\theta(z)}{d\theta} \, dz \bigg].
\end{aligned}
\tag{3}
$$

After the model is distilled into few-step form, it is converted into an autoregressive model by introducing causal attention. The conversion is carried out by sampling ODE trajectories from the teacher and training the autoregressive model on these trajectories. This stage functions as a warm-up phase, distinct from the main training procedure described below. The ODE training process is formally expressed in eq. (4).

$$
\mathcal{L}_{\text{ode}} = \mathbb{E}_{\mathbf{x},t} \bigg[ \Big\| G_\phi\Big( \{\mathbf{x}_{t_i}^{(i)}\}_{i=1}^N, \, \{t_i\}_{i=1}^N \Big) - \{\mathbf{x}_{\text{teacher}}^{(i)}\}_{i=1}^N \Big\|^2 \bigg].
\tag{4}
$$

## A.6 Improving Long-Term Smoothness via GRPO

Following the discussion in the main text, here we show the general form of GRPO which can be written as:

$$\mathcal{J}(\theta) = \mathbb{E}_{\{o_i\}_{i=1}^G \sim \pi_{\theta_{\text{old}}}(\cdot|c)} \mathbb{E}_{a_{t,i} \sim \pi_{\theta_{\text{old}}}(\cdot|s_{t,i})} \left[ \frac{1}{G} \sum_{i=1}^G \frac{1}{T} \sum_{t=1}^T \min\left( \rho_{t,i} A_i, \text{clip}(\rho_{t,i}, 1-\epsilon, 1+\epsilon) A_i \right) \right], \quad (5)$$

where $\rho_{t,i} = \frac{\pi_\theta(a_{t,i}\,|\,s_{t,i})}{\pi_{\theta_{\text{old}}}(a_{t,i}\,|\,s_{t,i})}$ is the importance weight, $\pi_\theta(a_{t,i}|s_{t,i})$ denotes the policy function for output $o_i$ at time step $t$ whose value can be computed according to eq. (1), $\epsilon$ is a clipping hyperparameter, and $A_i$ is the advantage computed across a generation group. The advantage is computed across a group of $G$ outputs as:

$$A_i = \frac{r_i - \text{mean}(\{r_1, r_2, \dots, r_G\})}{\text{std}(\{r_1, r_2, \dots, r_G\})}. \quad (6)$$

In order to generate adopt our model for GRPO, we inject Gaussian noise at each non-terminal step according to the noise scheduler. The probability of the final generated video can be formulated as below.

$$\begin{aligned}
\log p(x_{1:N}) &= \sum_{n=1}^N \log p(x_n \mid x_{<n}) \\
&= \sum_{n=1}^N \sum_{t=1}^T \sum_{i=1}^D \left[ -\frac{\left(x_{t,i}^{(n)} - (1-\sigma_t)\,x_{0,i}^{(n)}\right)^2}{2\sigma_t^2} - \log \sigma_t - \tfrac{1}{2}\log(2\pi) \right],
\end{aligned} \quad (7)$$

where $x_{0,i}^{(n)}$ is computed following eq. (1) conditioned on the previously generated samples $x_{<n}$, $D$ denotes the latent dimension size, $T$ is the number of non-terminal sampling steps, and $N$ is the total number of autoregressive steps.

## A.7 Temporal Repetition

As highlighted in RIFLEx (Zhao et al., 2025), one of the primary challenges in long video generation is *temporal repetition*, where videos begin to cycle with fixed, recurring patterns. In contrast, autoregressive approaches such as Self-Forcing (Huang et al., 2025) and our method, which rely exclusively on the KV cache to produce new frames, are less prone to this failure mode. To quantify this, we adopt the *NoRepeat Score* introduced in RIFLEx and report the results in table 8.

From table 8, we observe that NOVA, MAGI-1 and CausVid are more susceptible to repeated temporal patterns when extended to long videos. In contrast, methods that generate solely from the KV cache such as Self-Forcing and ours achieve stronger resistance to temporal repetition without requiring recomputation or overlapping frames.

Table 8: NoRepeat scores (↑) across different methods, computed following RIFLEx. The RIFLEx score reported here corresponds to its best published result and serves only as a reference.

| RIFLEx | Nova | MAGI-1 | SkyReels-V2 | CausVid | Self-Forcing | Ours |
|---|---|---|---|---|---|---|
| 89.0 | 67.19 | 73.44 | 95.31 | 92.97 | 100.0 | 98.44 |

## A.8 Evaluation with Gemini-2.5-Pro and Manually Verification

We present several representative results with Gemini-2.5-Pro on 50-second videos generated by our method and by baseline methods, as shown in figs. 8 and 9. None of the baselines sustain high quality at this length, and each displays distinct failure modes. CausVid consistently shows pronounced over-exposure even within its trained 5-second horizon, which worsens as the video progresses until motion collapses entirely. Self-Forcing suffers from severe error accumulation, leading to global darkening and stagnation. MAGI-1 initially avoids over-exposure, likely due to its reliance on diffusion forcing, but rapidly deteriorates into heavy over-exposure and structural collapse. SkyReels-V2, as seen in fig. 9, generally preserves structure but exhibits moderate to severe over-exposure, resembling CausVid's failure pattern.

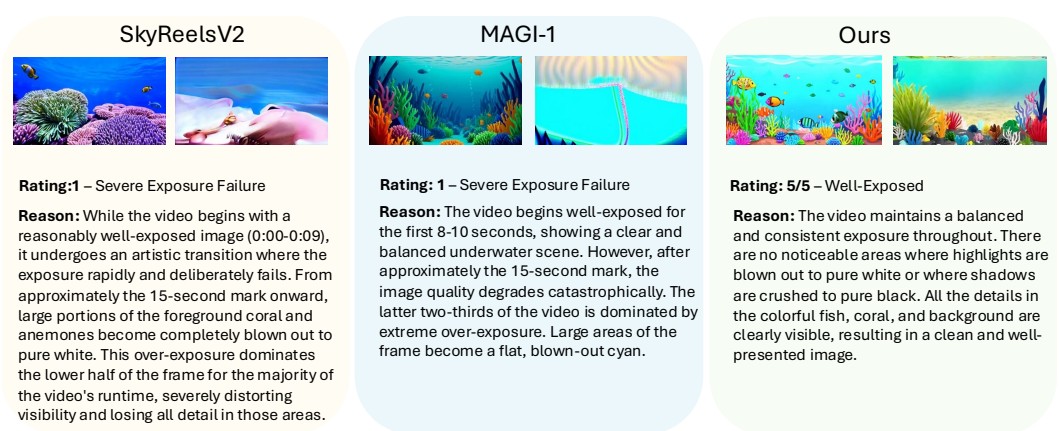

**CausVid**

**Rating: 2** – Noticeable Exposure Problems

**Reason:** The video consistently displays significant exposure issues, primarily with high contrast that leads to a loss of detail.

While the scene remains perfectly understandable, these "clear and persistent clipping in highlights or shadows" affect major portions of the frame (the main subject and its shadow), which aligns perfectly with the definition for a rating of 2.

**Self Forcing**

**Rating: 2** – Noticeable Exposure Problems

**Reason:** The large celestial body in the background is significantly overexposed, with large areas blown out to pure white, causing a complete loss of surface detail. The highlights on the astronaut's suit are also very bright and appear clipped.

These exposure issues are not fleeting; they are constant throughout the entire video.

**Ours**

**Rating: 5/5** – Well-Exposed

**Reason:** The video is well-exposed, demonstrating excellent handling of a high-contrast scene. The deep blacks of the space background and the harsh shadows on the lunar surface are appropriate for the setting and do not represent crushed blacks or loss of detail. Detail is well-preserved throughout the tonal range

Figure 8: Example evaluation using Gemini-2.5-pro on the results generated by our method, CausVid and Self-Forcing for prompt "An astronaut runs on the surface of the moon, the low angle shot shows the vast background of the moon, the movement is smooth and appears lightweight.". Gemini-2.5-Pro is tasked to rate the whole video with thinking and output reasoning first before outputting the final rating.

**SkyReelsV2**

**Rating:1** – Severe Exposure Failure

**Reason:** While the video begins with a reasonably well-exposed image (0:00-0:09), it undergoes an artistic transition where the exposure rapidly and deliberately fails. From approximately the 15-second mark onward, large portions of the foreground coral and anemones become completely blown out to pure white. This over-exposure dominates the lower half of the frame for the majority of the video's runtime, severely distorting visibility and losing all detail in those areas.

**MAGI-1**

**Rating: 1** – Severe Exposure Failure

**Reason:** The video begins well-exposed for the first 8-10 seconds, showing a clear and balanced underwater scene. However, after approximately the 15-second mark, the image quality degrades catastrophically. The latter two-thirds of the video is dominated by extreme over-exposure. Large areas of the frame become a flat, blown-out cyan.

**Ours**

**Rating: 5/5** – Well-Exposed

**Reason:** The video maintains a balanced and consistent exposure throughout. There are no noticeable areas where highlights are blown out to pure white or where shadows are crushed to pure black. All the details in the colorful fish, coral, and background are clearly visible, resulting in a clean and well-presented image.

Figure 9: Example evaluation using Gemini-2.5-pro on the results generated by our method, CausVid and Self-Forcing for prompt "A gorgeously rendered papercraft world of a coral reef, rife with colorful fish and sea creatures.". Gemini-2.5-Pro is tasked to rate the whole video with thinking and output reasoning first before outputting the final rating.

As further illustrated in figs. 8 and 9 and on our project page self-forcing-pp.github.io , all baselines demonstrate systematic breakdown in long-video generation that state-of-the-art MLLMs readily detect. To ensure alignment with human judgment, we conducted manual verification: 20 randomly sampled MovieGen videos were independently annotated by two authors, and the averaged scores were compared with Gemini-2.5-Pro. For 50-second sequences, Spearman's rank correlation reached 100% for the top three methods and 94.2% across all six baselines. Similar results are observed for the 75-second and 100-second videos, where the generation quality of baseline methods further declines.

Overall, our method achieves sustained long-term visual stability. Methods trained with diffusion forcing, such as SkyReels-V2 and MAGI-1, rank next, followed by CausVid, which maintains structure only under severe exposure. Both Self-Forcing and NOVA degrade to comparable low levels.

The prompt we use for evaluating is as following:

You are tasked with rating the exposure stability of a video. Assign a score according to the following scale:

0: Catastrophic Exposure. Nearly the entire frame is either blown out (pure white) or crushed (pure black), rendering the scene unreadable.

1: Severe Exposure Failure. Large portions of the frame are dominated by over-exposure or under-exposure, substantially impairing visibility.

2: Noticeable Exposure Problems. Persistent clipping is present in highlights or shadows. Significant areas lose detail, though the frame remains viewable.

3: Moderate Exposure Issues. Over-exposed highlights or under-exposed shadows occur but are limited in extent or duration.

4: Minor Exposure Flaws. Small regions are occasionally too bright or too dark, but these do not meaningfully disrupt overall visibility.

5: Well-Exposed. Balanced lighting across the frame. No distracting over-exposure or darkening; both highlights and shadows retain detail.

Do not claim that the observations in any video are of a specific artistic style or scene transitions unless the prompt explicitly states so. The prompt for generating the video is as follows:

First, provide a brief explanation of your reasoning, describing the observed exposure characteristics. Then, state your final score according to the scale.

## A.9 DISCUSSION OF DIFFUSION FORCING VS AUTOREGRESSIVE

Both diffusion forcing such as SkyReels and MAGI and autoregressive models with clean context such as Self-Forcing and ours can generate long videos. Diffusion forcing works by keeping a large number of frames in the current stage and apply different noise level for different frames. Thus, it naturally comes with better long term memory. However, such as long term memory comes at the cost of training instability as the number of different noise level combinations can be extremely huge due to the nature of diffusion models which needs multiple step denoising. Thus methods such as StreamDiT (Kodaira et al., 2025) has opted to distill the model first to limit the number of combinations which reduces the tranining instability. However, as the resutls shown in our work tables 1, 2 and 7 that a context with variable noises it not absolutely required to achieve long horizon generation with little quality degradation. As shown in our ablation study, the model is gradually learns to generate long videos with increased training budget even without using long video training set. We hope our work can help the community with generating better and more consistent long videos.

## A.10 MORE VISUALIZATIONS

Please checkout our demo page for more videos at https://self-forcing-plus-plus.github.io/

