# OpenReview forum: "Self-Forcing++: Towards Minute-Scale High-Quality Video Generation"
_ICLR.cc/2026/Conference — ICLR 2026 Poster_

### Official Review · Reviewer_hYZq · 2025-10-26

**Soundness:** 3
**Presentation:** 4
**Contribution:** 3
**Rating:** 8
**Confidence:** 3

**Summary:**

The paper introduces Self-Forcing++, a training framework for minute-scale autoregressive video generation that mitigates long-horizon degradation without long-video teachers or datasets. It identifies a dual train–test mismatch: models train with dense teacher supervision on short clips but must sustain much longer rollouts at inference, causing compounding errors (motion stalling, exposure drift).
Self-Forcing++ rolls the student out far beyond the teacher’s horizon, re-injects diffusion noise (Backward Noise Initialization) to preserve temporal context, and performs Extended Distribution Matching Distillation by uniformly sampling K-length windows (teacher horizon) from the long rollout for student–teacher alignment. Training consistently uses a rolling KV cache, eliminating cache mismatches and overlapping-frame recomputation. An optional GRPO stage with an optical-flow smoothness reward further enhances temporal continuity.
Experiments show substantial gains over CausVid, Self-Forcing, SkyReels-V2, and MAGI-1 on 50/75/100s tasks, especially in motion dynamics and a proposed Visual Stability metric that addresses VBench’s long-video bias. With increased compute, the model scales to 4m15s while maintaining quality.

**Strengths:**

1.	The paper clearly diagnoses the core barrier to long-horizon video generation as a dual train–inference mismatch in temporal horizon and supervision density. The overall pipeline is communicated cleanly, with a step-by-step flow (long self-rollout, backward noise re-injection, sliding-window distillation, rolling KV cache). The failure modes of prior work, including motion collapse, exposure drift, and cache-state mismatches, are explained convincingly and tied to design choices.
2.	The method achieves strong, consistent gains over baselines (CausVid, Self-Forcing, SkyReels-V2, MAGI-1) on comparable 50–100s tasks, especially in motion dynamics and exposure/visual stability. With increased training budget, and without long-video data or teachers, it scales to 4m15s (near positional embedding limits) while maintaining coherent motion and stable exposure.
3.	The authors identify and analyze a bias in VBench that can favor over-exposed or degraded frames, which are common failure modes in prior systems. They propose Visual Stability, an evaluation protocol leveraging a strong video MLLM and human verification to better capture long-horizon degradation and exposure issues, thereby improving the reliability of benchmarking for long video generation.

**Weaknesses:**

1.	The paper should provide a clearer accounting of computational resources for both training and inference, such as FLOPs, GPU hours, and memory, to enable fair cost–quality comparisons with baselines.
2.	While the paper includes ablations (attention window, noisy-KV, GRPO), it would be even stronger with finer analyses of key design choices, e.g., alternative window sampling distributions, different noise schedules/intensities for backward initialization, and systematic sweeps over K/N ratios.
3.	There are minor punctuation and typography issues. For example, in the Introduction’s first paragraph the last sentence ends with a comma instead of a period, and around line 202 the phrase "by $\Theta$" is followed by an extra period. A careful proofreading pass to fix such punctuation/formatting inconsistencies would improve presentation quality.

**Questions:**

-

---

> ### Author Response · Authors · 2025-11-21
> **thank you for your review**
>
> Thank the reviewer for recognizing our contribution. We truly appreciate the reviewer’s valuable feedback and suggestions. Please see our responses below.
>
> **Q1**: The paper should provide a clearer accounting of computational resources for both training and inference, such as FLOPs, GPU hours, and memory, to enable fair cost–quality comparisons with baselines.
>
> **A1**: Thank the reviewer for the suggestion. Our main experiment is trained with a total of 48 H100 GPU days for full training with 80GB memory. The training cost could be further reduced with LoRA [1]. For inference, it takes around 50 seconds to generate a 50-second video which occupies around 22GB of memory on one H100 GPU with a total of 80GB memory. This is the same as our baseline method Self-Forcing. We have updated our writing to include these details in our writing.
>
> [1] Hu, Edward J., et al. "Lora: Low-rank adaptation of large language models." ICLR 1.2 (2022): 3.

---

> > ### Author Response · Authors · 2025-11-21
> >
> > **Q2**: While the paper includes ablations (attention window, noisy-KV, GRPO), it would be even stronger with finer analyses of key design choices, e.g., alternative window sampling distributions, different noise schedules/intensities for backward initialization, and systematic sweeps over K/N ratios.
> >
> > **A2**: Thank you so much for your feedback. Following the reviewer’s suggestion, we have added the following 3 ablation studies to further analyze our design choices.
> >
> > 1. Our main experiment utilizes a uniform window sampling distribution, here we utilize a beta sampling to bias the sampled window toward more shorter-sequences.  In summary, the trained model still outperforms the baseline method significantly, however the motion has reduced compared to uniform sampling likely due to not enough supervision towards later part of the generation sequences.
> > 2. Our main experiment enables gradient computation only on the final noise schedule of the 4-step generator. Following the reviewer’s suggestion, we evaluate a variant that enables gradients uniformly across all noise schedules. Overall, allowing gradients at all steps imposes stronger alignment with the teacher, yielding higher stability, but at the cost of substantially reduced motion dynamics. In contrast, our method preserves more natural motion while maintaining comparable stability and generation quality.
> > 3. Because the teacher is trained on 5-second videos, we set K=5 seconds (N=21 latent frames) and add a cost-efficient experiment using a 10-second window (N=42 latent frames). This extension reduces flickering and darkening but does not solve long-sequence error accumulation. It’s likely that the model still lacks exposure to sufficiently degraded long roll-outs during training, preventing it from learning effective long-range error correction. Following on the reviewer’s question 1, we plan to employ LoRA fine-tuning to reduce training cost and perform more systematic sweeps over K/N ratios as our following work.
> >
> > We have also included these results into our ablation study section. We will add more results with different parameters as the reviewer suggested in the final version of the paper. Thank the reviewer again for the great suggestions.
> >
> > [1] Huang, Xun, et al. "Self Forcing: Bridging the Train-Test Gap in Autoregressive Video Diffusion." arXiv preprint arXiv:2506.08009 (2025).

---

> > > ### Author Response · Authors · 2025-11-21
> > >
> > > **Q3**: There are minor punctuation and typography issues. For example, in the Introduction’s first paragraph the last sentence ends with a comma instead of a period, and around line 202 the phrase "by " is followed by an extra period. A careful proofreading pass to fix such punctuation/formatting inconsistencies would improve presentation quality.
> > >
> > >
> > > **A3**: Thank the reviewer for the careful review. We have modified our manuscript to fix the issues. We will conduct more proofreading passes to improve our presentation quality. Thank you again for your time.

---

> ### Author Response · Authors · 2025-11-21
>
> We sincerely thank the reviewer again for your time reviewing our paper and providing valuable feedback. We truly appreciate it.

---

### Official Review · Reviewer_PrTo · 2025-10-28

**Soundness:** 3
**Presentation:** 3
**Contribution:** 3
**Rating:** 8
**Confidence:** 4

**Summary:**

This paper introduces Self-Forcing++, a method that achieves a landmark breakthrough in the stability of long-horizon autoregressive video generation. It directly confronts the critical issue of quality degradation that occurs when models distilled from short-video teachers are tasked with generating minute-scale sequences. The core contribution is an elegant and highly effective training strategy that requires no long-video data. The method involves prompting the student model to generate long, error-accumulated rollouts, and then using the short-horizon teacher to perform "segment-wise correction" on randomly sampled clips from these rollouts. This process effectively teaches the model to recover from its own extrapolation errors.
Empirically, the results are groundbreaking. The method extends high-fidelity video generation from a baseline of ~10 seconds to over four minutes, a feat previously considered out of reach. It demonstrates an unprecedented ability to maintain local visual quality, motion dynamics, and temporal coherence over these extended durations.

**Strengths:**

1. The paper's most significant and undeniable contribution is its success in mechanically extending the stable generation horizon of autoregressive models by an order of magnitude. The ability to produce minute-scale videos without collapsing is a remarkable engineering feat that directly addresses a major bottleneck in the field. This establishes a new, albeit signal-level, state-of-the-art for generation duration.
2. The conceptual reframing of the long-video problem is the paper's most elegant intellectual contribution. The core insight—that a short-horizon teacher can be repurposed as a "local error corrector" for a student's long, degraded rollouts—is a highly non-obvious and clever idea. This formulation provides a novel, data-efficient paradigm for tackling extrapolation challenges.
3. A key practical finding of this work is the demonstrated correlation between the training budget and the achievable horizon of stable generation (Figure 6). This provides the community with a clear, albeit potentially brute-force, recipe for progress: longer, stable videos can be achieved by investing more computation into the self-correction training loop. This is a valuable and actionable result.
4. The authors should be credited for their critical examination of existing evaluation protocols. Identifying and attempting to remedy the biases of VBench when applied to long-form video demonstrates a level of scientific rigor and thoughtfulness that is often missing. While their proposed metric is deeply flawed, the motivation to improve evaluation standards is in itself a positive contribution to the community's discourse.

**Weaknesses:**

1. Architectural Constraint: A Bounded Information Horizon via the KV Cache.
The model's autoregressive generation relies on a fixed-size rolling KV cache of length L. This mechanism architecturally imposes a strict Markov assumption on the generative process. The conditional probability of generating the current latent state x_t is not conditioned on the entire history x_{<t}, but is instead approximated by p(x_t | x_{t-L:t-1}). Consequently, the model's information horizon is strictly bounded by L. Any semantic dependency or causal relationship requiring the integration of information over a temporal span greater than L (e.g., long-term entity consistency, narrative causality) cannot be modeled, as the requisite information has been discarded from the context window. This is a fundamental architectural limitation, not a matter of training scale.
2. Objective Function Constraint: Optimization of Local, not Global, Properties.
The proposed Self-Forcing++ training objective is formulated as an expectation over uniformly sampled, short, contiguous segments of length K (where K is the teacher's horizon). The loss function is a sum of local distribution matching objectives, each minimizing the KL divergence between the student's and teacher's distributions within that local window: KL(p_θ(x_i:i+K) || p_T(x_i:i+K)). This objective function provides a powerful supervisory signal for enforcing local properties—such as texture fidelity, motion dynamics, and exposure consistency—that are observable within a segment of length K. However, it provides no direct supervisory signal for dependencies between non-overlapping segments, W_i and W_j, where j > i+K. The model is therefore trained to be an expert local generator, but it is not explicitly optimized for global semantic coherence. The observed long-term stability is an emergent property of chaining together high-fidelity local segments, not a result of learning a true global model of the video distribution.

**Questions:**

1. On the Inherent Trade-offs of the Global-to-Local Distillation: Your approach involves distilling knowledge from a bidirectional, full-attention teacher into a causal, autoregressive student. This global-to-local translation necessarily involves trade-offs. Could you comment on what capabilities are inherently lost in this process? For example, a bidirectional model can generate "perfectly looping" videos or effects where the beginning is influenced by the end. Are such temporally non-causal phenomena fundamentally beyond the reach of your framework, and do you see this as a necessary price for achieving scalability and streaming capabilities?

---

> ### Author Response · Authors · 2025-11-21
> **thank you for your review**
>
> We sincerely thank the reviewer for recognizing our contribution and providing insightful feedback. We really appreciate it. Please see our responses to your questions below.
>
> **Q1**: Architectural Constraint: A Bounded Information Horizon via the KV Cache. The model's autoregressive generation relies on a fixed-size rolling KV cache of length L. This mechanism architecturally imposes a strict Markov assumption on the generative process. The conditional probability of generating the current latent state x_t is not conditioned on the entire history x_{<t}, but is instead approximated by p(x_t | x_{t-L:t-1}). Consequently, the model's information horizon is strictly bounded by L. Any semantic dependency or causal relationship requiring the integration of information over a temporal span greater than L (e.g., long-term entity consistency, narrative causality) cannot be modeled, as the requisite information has been discarded from the context window. This is a fundamental architectural limitation, not a matter of training scale.
>
> **A1**: Thank the reviewer for the feedback. We agree with the reviewer that due to the large amount of vision tokens, local-attention or sparse attention is usually utilized to get around this limitation. Addressing long term memories remains one of our top future work, and it also aligns closely with the research communities’ focus on advancing long-video generation. We are currently exploring techniques such as compressing the history tokens similar to FramePack [1], hierarchical memory strategy or state space models which was recently adopted by [2] for longer memory preservation.
>
> **Q2**: Objective Function Constraint: Optimization of Local, not Global, Properties. The proposed Self-Forcing++ training objective is formulated as an expectation over uniformly sampled, short, contiguous segments of length K (where K is the teacher's horizon). The loss function is a sum of local distribution matching objectives, each minimizing the KL divergence between the student's and teacher's distributions within that local window: KL(p_θ(x_i:i+K) || p_T(x_i:i+K)). This objective function provides a powerful supervisory signal for enforcing local properties—such as texture fidelity, motion dynamics, and exposure consistency—that are observable within a segment of length K. However, it provides no direct supervisory signal for dependencies between non-overlapping segments, W_i and W_j, where j > i+K. The model is therefore trained to be an expert local generator, but it is not explicitly optimized for global semantic coherence. The observed long-term stability is an emergent property of chaining together high-fidelity local segments, not a result of learning a true global model of the video distribution.
>
> **A2**: Thank the reviewer for the insightful feedback. Due to the scarcity of high quality long videos, our method provides a data-free approach to alleviate error accumulation in long form video generation. As the reviewer suggested, in order to achieve a true global model, we need a large-scale high quality long video dataset or a stronger teacher model to provide more powerful supervisory signals so that our approximation is closer to a true global distribution. We will iterate on our current method to further improve the generation quality.
>
>
> [1] Zhang, Lvmin, et al. "Frame Context Packing and Drift Prevention in Next-Frame-Prediction Video Diffusion Models." The Thirty-ninth Annual Conference on Neural Information Processing Systems.
> [2] Shin, Joonghyuk, et al. "MotionStream: Real-Time Video Generation with Interactive Motion Controls." arXiv preprint arXiv:2511.01266 (2025).

---

> > ### Author Response · Authors · 2025-11-21
> >
> > **Q3**: On the Inherent Trade-offs of the Global-to-Local Distillation: Your approach involves distilling knowledge from a bidirectional, full-attention teacher into a causal, autoregressive student. This global-to-local translation necessarily involves trade-offs. Could you comment on what capabilities are inherently lost in this process? For example, a bidirectional model can generate "perfectly looping" videos or effects where the beginning is influenced by the end. Are such temporally non-causal phenomena fundamentally beyond the reach of your framework, and do you see this as a necessary price for achieving scalability and streaming capabilities?
> >
> >
> > **A3**: Thank you so much for your great insight. We totally agree with the reviewer on this. Bidirectional models have the best control over the whole generation span and are able to deal with much more complex scenes which are currently hard to achieve by autoregressive models with local attention. Besides the example given by the reviewer, we also notice phenomena such as objects abruptly emerging or disappearing due to the lack of overall knowledge which can be alleviated to some extent with post-training methods as described in our section 3.3. In order to fully address them, the following techniques can be incorporated into our current framework to greatly enhance its ability
> >
> > 1. **Long term memory** which corresponds to the reviewer’s Q1 above can be readily incorporated into our current framework. With long term memory, the model will possess much more information to better generate next frames autoregressively. E.g, in the case of generating a perfect loop, the model knows its starting state and the overall status, thus it will better handle such cases.
> > 2. **Variable trunk size**. Our autoregressive model is trained with a fixed trunk size of 3. However, variable trunk size generation is supported by our current framework as well because trunk size is configurable. Therefore, in places where global control is more needed such as the scenario described by the reviewer, a larger trunk size can be utilized, in places where global control is less needed, a small trunk size can be used. However, using a larger trunk size will increase the computation which incurs reduced FPS. How to perform generation more efficiently is another direction we plan to explore in our future work.
> > 3. **Global prompt alignment**. In order to handle complex scenes, autoregressive models normally utilize another LLM to divide complex scenes into subprompts and apply them to autoregressive models sequentially. However, as the reviewer mentioned above, the model will lose the ability to know what will happen after that which is available in bi-directional generations. We are thinking about feeding all the prompts to the autoregressive model altogether, and apply techniques similar to temporal position embedding to prompts as well so that the model not only knows the prompt it is generating, but also knows the whole picture.
> >
> > Overall, we think the gap between autoregressive models and bi-directional models will be greatly reduced with the above improvements.

---

> > > ### Author Response · Authors · 2025-11-21
> > >
> > > We truly thank the reviewer for the insightful questions and suggestions which are the fundamental problems in the field of long video generation. What the reviewer suggested remain the top priorities of our next step such as addressing long term memory, further improving the generation quality and recovering the capabilities that are lost during the transformation. We really appreciate all these suggestions and will actively work on them.

---

### Official Review · Reviewer_avME · 2025-11-02

**Soundness:** 3
**Presentation:** 2
**Contribution:** 3
**Rating:** 6
**Confidence:** 5

**Summary:**

The paper proposes an algorithm to improve the length extrapolation capability of Self-Forcing-style autoregressive video diffusion models. In Self Forcing, due to the limitation that the teacher model is only trained on short duration, the student perform short-duration rollout (e.g. 5s) and receives supervision from the teacher trained on the same length. Here, the student model performs extended-duration rollout

**Strengths:**

- The paper demonstrates strong empirical results with multi-minute-long generation with no significant error accumulation. This is achieved without supervision from long videos by leveraging teacher feedback from short individual video segments generating by the student via self-rollout.
- The experiments are very comprehensive. The paper not only demonstrates the advantage over prior work (e.g., Self Forcing) but also compares with alternative strategies (noisy kv or local attention) that alleviate error accumulation. The paper also identifies an important issue in existing benchmark (VBench) and proposes a better benchmarking.

**Weaknesses:**

- The description of "Backwards Noise Initialization" in Section 3.2 is very unclear and confusing.
  - Is the "Backwards Noise Initialization" term comes from "backward simulation" in DMD2? It seems to be the case when I read the sentence "they used this for... circumvent the need for real training data". But in Fig. 2, it seems to refer to injecting noise to student output before computing fake/real scores. The noise injection is employed in the original DMD and all follow-up works, is unrelated to "circumvent real training data", and should not be an original contribution.
  - Does t represent noise level or frame index? Sometime it seems to refer to noise level (x_t = (1 − σ_t)x_0 + σ_tϵ) but sometime it seems to refer to the frame index (N clean frames and clean trajectory {x^S_t}_{t=1}^N)
  - Is there still a fake score network with separate parameters, like in DMD? If so, in eq. 2 the fake score s^S_\theta should not be denoted with parameter theta - theta is the parameter of the student.
  - What is S in x_t^S (L198)?
- The way Figure 2 compares CausVid/SF/SF++ is confusing. The second row of frames of CausVid (frames with independent noise levels) are the input to the **student** model. But the second row of frames of the SF/SF++ appears to be the input to the **teacher** model and the fake score network. The input of the student model for SF/SF++ is not shown and the output of the student model is illustrated in the first row. CausVid should also have the corresponding "Backward Noise Initialization" step but it's subsumed in the "DMD Alignment" process. In short, the frames depicted in the same row, which are supposed to be side-to-side comparisons, do not correspond to things at the same stage.
- Would be great if the authors can discuss the relationship with APT2 (Lin et al. 2025), which also uses a similar long video training method to improve long video generation ability.

**Questions:**

- What is the exact thing that is scaled in the "Training Budget Scaling" section (4.4)? Is it the batch size, number of iterations, or per-iteration rollout-duration. Could you provide detailed experiment config for each (1x, 4x, 8x, 20x, 25x) training setup?
- Have you compared the proposed strategy (rollout an entire, long video then uniformly sample a single segment, e.g., rollout 1 min then sample 5s) versus an alternative strategy that rollout one segment at a time, compute loss for it and then rollout another segment (e.g., rollout 5s, compute loss and backprop, then rollout 5s).

---

> ### Author Response · Authors · 2025-11-21
> **thank you for your review**
>
> Thank you so much for your time reviewing our paper and providing valuable feedback. We sincerely appreciate it. Please see our responses below.
>
> **Q1**: Is the "Backwards Noise Initialization" term comes from "backward simulation" in DMD2? It seems to be the case when I read the sentence "they used this for... circumvent the need for real training data". But in Fig. 2, it seems to refer to injecting noise to student output before computing fake/real scores. The noise injection is employed in the original DMD and all follow-up works, is unrelated to "circumvent real training data", and should not be an original contribution.
>
> **A1**: Sorry for the confusion, it actually refers to both of them. The key problem we are trying to explain in this paragraph is why cross segment consistency can be achieved even if the teacher is a short-horizon generation model. In APT2 [1] (we further discuss in Q6) and LongLive [2] (we further discuss in Q8), they both use an overlapping evaluation strategy to encourage cross segment continuation, which is not used in our method (for each generation, we just sample one window as the reviewer described in Q8). The cross-segment consistency in our method mainly comes from the following 2 aspects:
>
> 1. The clean latents are generated by the model autoregressively, it offers 2 advantages, 1) no real long training dataset is needed any more which is similar to “backwards simulation” in DMD2 2)  The starting clean latents (although with error accumulation) is already consistent based on our observations as mentioned in our motivation paragraph.
>
> 2. Based on point 2) above, if we apply DMD to these already continuous latents, the teacher will be able to give supervision signals that remain consistent with previous generations. The point we are trying to make here is not that noise injection helps with computing real and fake scores as the original DMD mentioned by the reviewer above but more of that they operate on starting latents where noises are added back to continuous latents.
>
> So we give it a different name other than using “backward simulation”. If the reviewer finds the name confusing, we will change it to a different name.
>
> [1] Lin, Shanchuan, et al. "Autoregressive Adversarial Post-Training for Real-Time Interactive Video Generation." arXiv preprint arXiv:2506.09350 (2025).
> [2] Yang, Shuai, et al. "Longlive: Real-time interactive long video generation." arXiv preprint arXiv:2509.22622 (2025).

---

> > ### Author Response · Authors · 2025-11-21
> >
> > **Q2**: Does t represent noise level or frame index? Sometime it seems to refer to noise level (x_t = (1 − σ_t)x_0 + σ_tϵ) but sometime it seems to refer to the frame index (N clean frames and clean trajectory {x^S_t}_{t=1}^N)
> >
> > **A2**: Thank you for your careful review. We should have used a different notation for noise level and frame index. We have modified our manuscript to fix it.
> >
> > **Q3**: Is there still a fake score network with separate parameters, like in DMD? If so, in eq. 2 the fake score s^S_\theta should not be denoted with parameter theta - theta is the parameter of the student.
> >
> > **A3**: Yes, we extended the original DMD into a windowed DMD but still kept its original form. Thank you for your suggestion, we have removed it from eq2 in our writing.
> >
> > **Q4**: What is S in x_t^S (L198)?
> >
> > **A4**: S indicates that the trajectory is generated by the student model. We removed it from our writing to not cause confusion since it’s redundant with the text after it. Thank you!
> >
> > **Q5**: The way Figure 2 compares CausVid/SF/SF++ is confusing. The second row of frames of CausVid (frames with independent noise levels) are the input to the student model. But the second row of frames of the SF/SF++ appears to be the input to the teacher model and the fake score network. The input of the student model for SF/SF++ is not shown and the output of the student model is illustrated in the first row. CausVid should also have the corresponding "Backward Noise Initialization" step but it's subsumed in the "DMD Alignment" process. In short, the frames depicted in the same row, which are supposed to be side-to-side comparisons, do not correspond to things at the same stage.
> >
> > **A5**: Thank you so much for your suggestion. We drew it this way mainly to highlight the difference in the training paradigms of CausVid where CausVid adopts independent noise levels which are not utilized in Self-Foricing and ours (second row) and how temporal consistency is achieved by using the self-generated sequences in contrast to the real dataset used by Causvid (first row). Sorry for the confusion, we will iterate on it to find a better way to illustrate the differences.

---

> ### Author Response · Authors · 2025-11-21
>
> **Q6**: Would be great if the authors can discuss the relationship with APT2 (Lin et al. 2025), which also uses a similar long video training method to improve long video generation ability.
>
> **A6**: Thank you for the suggestion, we have updated our writing to discuss the relationship with APT2 in our related work. In summary, APT2 [1] adopts an adversarial training paradigm that first transforms a bidirectional model into a one-step autoregressive generator, then enhances generation quality through joint training with student forcing and a discriminator trained on both real and generated video segments. However, our method relies on pretrained diffusion models and does not require a real training dataset and the training of a discriminator.  The overall workflow for training long video generation is similar to ours, but with a different training procedure similar to what the reviewer mentioned in Q8 below.
>
> **Q7**: What is the exact thing that is scaled in the "Training Budget Scaling" section (4.4)? Is it the batch size, number of iterations, or per-iteration rollout-duration. Could you provide detailed experiment config for each (1x, 4x, 8x, 20x, 25x) training setup?
>
> **A7**: Sorry for the confusion. During our development, we noticed that both scaling up rollout-duration and training iteration results in improved long-term generation quality. In this experiment, we mainly scaled up the number of iterations to see if error-accumulation can be further reduced when extending to ultra long sequences. The config we used is training with 8 H100 GPUs with a batch size of 8 and same optimization hyperparameters as the main experiments. The rollout-duration is 50 seconds. It takes around 500 iterations for the model to stably generate 5-second videos from ODE initialized checkpoint, the final demo shown in our section 4.4 is generated at iteration 12500.  We have included these details into our paper.
>
>
> [1] Lin, Shanchuan, et al. "Autoregressive Adversarial Post-Training for Real-Time Interactive Video Generation." arXiv preprint arXiv:2506.09350 (2025).

---

> > ### Author Response · Authors · 2025-11-21
> >
> > **Q8**: Have you compared the proposed strategy (rollout an entire, long video then uniformly sample a single segment, e.g., rollout 1 min then sample 5s) versus an alternative strategy that rollout one segment at a time, compute loss for it and then rollout another segment (e.g., rollout 5s, compute loss and backprop, then rollout 5s).
> >
> > **A8**: Thank the reviewer for the great suggestion. Yes, we have tried it during development.
> > There is a concurrent work LongLive [1] (released after the ICLR submission deadline) which adopts exactly the same idea as the reviewer’s suggestion. Although the idea is similar to ours, there is a key difference between our method and LongLive.
> >
> > LongLive can be thought of as tackling the problem from a distillation perspective as the reviewer suggested. However, since the starting frame is an image frame without temporal compression, it possesses a different distribution as other frames which requires special handling to avoid flickering as shown in Self-Forcing [2] (section 3.4), a phenomenon we also observe during our development. Thus, during training, LongLive has to constantly decode the generated videos back to pixel space and reencode the first frame.
> >
> > Different from LongLive, we utilize the teacher model similar to a latent reward model, the flickering problem naturally disappears as training goes on as demonstrated in our section 4.4 without special handling of any frames. Therefore, our method operates entirely in the latent space. We attribute this to the smoothing effect of windowed sampling (rollout first then sample as described by the reviewer). As a result, when scaling to larger models or operating under constrained memory, our method is more memory-efficient when loading the VAE and decoding into pixel space is memory-consuming.
> >
> > Overall, besides the distillation perspective adopted by our concurrent work LongLive [1],  our method provides a different approach from the reward perspective in addressing long-horizon error accumulation problems.
> >
> > [1] Yang, Shuai, et al. "Longlive: Real-time interactive long video generation." arXiv preprint arXiv:2509.22622 (2025).
> > [2] Huang, Xun, et al. "Self Forcing: Bridging the Train-Test Gap in Autoregressive Video Diffusion." arXiv preprint arXiv:2506.08009 (2025).

---

> > > ### Author Response · Authors · 2025-11-21
> > >
> > > We sincerely thank the reviewer again for your insightful suggestions and feedback, we really appreciate it!

---

### Author Response · Authors · 2025-12-03
**Final Rebuttal Summary**

Dear Reviewers and AC,

As the rebuttal period is ending, we want to express our gratitude again to you for your time reviewing our work, recognizing our contributions and sharing insightful feedback and suggestions. We really appreciate it. Here is a summary of the reviewers’ main questions and our responses.

1. For reviewer **avME**, we mainly addressed
- the confusion regarding the **terminology** “backwards noise initialization” and figure 2 plot where we used it to refer to both “backwards noise simulation” and adding noise to compute distribution matching, which is  the key technique we used to achieve long term consistency even with windowed DMD.
- The major difference between our window-sampling based training regime and gradual roll-out training strategy where unlike concurrent works, our method **solely operates in the latent space** and doesn’t need to constantly decode back into pixel spaces. Thus our method is more memory-efficient.
2. For reviewer **PrTo**, we mainly discussed the architectural constraints of the current approach and how such **gaps** such as long term memory can be addressed within our framework in future works, which highly aligns with the top priorities of the research community.
3. For reviewer **hYZq**, besides the ablation studies in the paper, we added **3 more ablation studies** suggested by the reviewer covering different aspects such as sampling strategy, different noise schedules and K/N ratios.


We have also included more detailed training budgets as suggested by reviewers (reviewer avME, hYZq) and corrected the notation and typography issues (reviewer avME, hYZq). We have **updated our manuscript** to reflect all these changes (marked in blue text).



We sincerely thank all of the reviewers again for recognizing the strength of our proposed method where **(1)** we extend current long video generation by **an order of magnitude** with no significant error accumulation while solely relying on a pre-trained short-horizon teacher, which were considered out of reach in the field (reviewer avME, PrTo, hYZq), **(2)** the identification of **flaws** in popularly used video benchmarks (reviewer avME, PrTo, hYZq), **(3)** the **comprehensiveness** of our experiments (reviewer avME, hYZq), and **(4)** the demonstration of the **correlation** between training budget and stable long horizon generation which provides a viable way for the community to effectively address error-accumulations in ultra long videos generation (reviewer PrTo, hYZq).

**We truly appreciate it.**

Paper 4671 Authors

---

### Meta-Review · Area_Chair_Vq2o · 2025-12-29

**Summary:**

The initial reviews highlighted several shared concerns, focusing mainly on the need for technical clarification regarding model components (e.g., "Backwards Noise Initialization"), missing empirical details (computational resources and finer ablations), and fundamental theoretical constraints (e.g., the bounded information horizon). During the rebuttal phase, the authors successfully clarified the methodology, added comprehensive computational details and three key ablation studies, and provided a thoughtful discussion on the inherent architectural limitations. The responses were strong and resolved most major concerns, leading to a robust consensus. Given the general agreement that the approach is solid and the results are promising, I recommend Accept (Poster).

**Reviewer Concerns:**

Addressed during Rebuttal:

- Technical Ambiguity (avME): The authors clarified the confusing term "Backwards Noise Initialization," fixed notation inconsistencies, and promised to iterate on the confusing Figure 2. The relationship with APT2 was also successfully discussed.

- Computational Details (hYZq): Comprehensive training (48 H100 GPU days) and inference cost details were added, enabling fair comparisons.

- Finer Ablations (hYZq): The authors added three key ablation studies: alternative window sampling (beta-sampling), gradient computation (uniform vs. final step), and a larger window size ($K=10$s), significantly strengthening the empirical analysis.

Outstanding/Minor Points:

- Fundamental Architectural and Objective Limitations (PrTo): The reviewer raised two fundamental constraints: the bounded information horizon due to the fixed KV cache ($L$) and the objective's focus on local segment optimization ($K$). The authors acknowledged these as known, structural limitations of the current paradigm and committed to exploring them in future work. This is not considered a major weakness against acceptance.

**Reviewer Scores:**

Reviewer avME (Rating: 6 $\to$ 6): The primary technical and clarity concerns were addressed (e.g., "Backwards Noise Initialization," notation, Figure 2). This resolves the major confusion points, warranting the accept decision.

Reviewer PrTo (Rating: 8 $\to$ 8): The reviewer's concerns were insightful critiques of fundamental, structural limitations. Since the initial rating was already high and the authors provided thoughtful acknowledgment, the score is likely to remain stable.

Reviewer hYZq (Rating: 8 $\to$ 8): The reviewer's primary requests—clearer computational accounting and finer ablation studies—were fully and successfully addressed. This solid response reinforces the initial positive assessment, leading to a stable score.

---

### Decision · Program_Chairs · 2026-01-26

Accept (Poster)